# Status of Wheat Rust Research and Progress in Rust Management-Indian Context

**Subhash C. Bhardwaj** [1,*], **Gyanendra P. Singh** [2], **Om P. Gangwar** [1], **Pramod Prasad** [1] and **Subodh Kumar** [1]

1   ICAR-Indian Institute of Wheat and Barley Research, Flowerdale, Shimla 171002, Himachal Pradesh, India; gangwarop@gmail.com (O.P.G.); pramoddewli@gmail.com (P.P.); subodhfdl@gmail.com (S.K.)
2   ICAR-Indian Institute of Wheat and Barley Research, Karnal 132001, Haryana, India; director.iiwbr@icar.gov.in
*   Correspondence: scbfdl09@gmail.com

**Abstract:** The rusts of wheat, caused by three species of *Puccinia*, are very devastating diseases and are major biotic constraints in efforts to sustain wheat production worldwide. Their capacity to spread aerially over long distances, rapid production of infectious uredospores, and abilities to evolve new pathotypes, makes the management of wheat pathogens a very challenging task. The development and deployment of resistant wheat varieties has proven to be the most economic, effective and efficient means of managing rust diseases. Rust resistance used in wheat improvement has included sources from the primary gene pool as well as from species distantly related to wheat. The 1BL/1RS translocation from cereal rye was used widely in wheat breeding, and for some time provided resistance to the wheat leaf rust, stripe rust, and stem rust pathogens conferred by genes *Lr26*, *Yr9*, and *Sr31*, respectively. However, the emergence of virulence for all three genes, and stripe rust resistance gene *Yr27*, has posed major threats to the cultivation of wheat globally. To overcome this threat, efforts are going on worldwide to monitor rust diseases, identify rust pathotypes, and to evaluate wheat germplasm for rust resistance. Anticipatory breeding and the responsible deployment of rust resistant cultivars have proven to be effective strategies to manage wheat rusts. Efforts are still however being made to decipher the recurrence of wheat rusts, their epidemiologies, and new genomic approaches are being used to break the yield barriers and manage biotic stresses such as the rusts. Efficient monitoring of pathotypes of *Puccinia* species on wheat, identification of resistance sources, pre-emptive breeding, and strategic deployment of rust resistant wheat cultivars have been the key factors to effective management of wheat rusts in India. The success in containing wheat rusts in India can be gauged by the fact that we had no wheat rust epiphytotic for nearly last five decades. This publication provides a comprehensive overview of the wheat rust research conducted in India.

**Keywords:** rust management; wheat; resistance; epidemiology; monitoring; pathotype; *Puccinia*

## 1. Introduction

Globally, wheat is cultivated on an area of about 219 million hectares with a production of 763.2 million tons. With the burgeoning population, requirement for food will increase; India alone will need more than 140 million tons of wheat by 2050 to feed an estimated population of 1.73 billion [1]. With the intensification of agriculture, pest and pathogen incidences are increasing. New biotic threats like Ug99 of stem rust, virulence for resistance gene *Yr9* in the stripe rust pathogen, and the new threat of wheat blast have emerged. To sustain and increase the wheat production, reducing the impact of biotic and abiotic stresses on production will be critical.

Though many biotic stresses impede wheat production, the wheat rust diseases are best known for their devastating and widespread nature. Both stem rust and stripe rust can cause 100% loss, whereas leaf rust can result in 50% loss [2]. In India stripe rust of wheat (caused by *Puccinia striiformis* f. sp. *tritici* Westend., Authority) is a threat in 10 million hectares of Northern India, whereas stem rust (caused by *P. graminis* Pers. f. sp. *tritici* Eriks. & Henn.,) threatens about 7 million hectares of Central and Peninsular India. In contrast, leaf rust (caused by *P. triticina* Eriks. Authority) is prevalent wherever wheat is grown (Figure 1). In India, about 10 million tons of wheat is saved every year through the successful management of wheat rusts. However, the rust pathogens continue to affect production through the occurrence of new virulences that have led to the discontinuance of important wheat cultivars. For example, the emergence of virulence for *Yr9* in *P. striiformis* in India led to the elimination of PBW343, a mega variety. Such changes in the virulence patterns of wheat rust pathogens continuously remind us of the threat they pose to global wheat production.

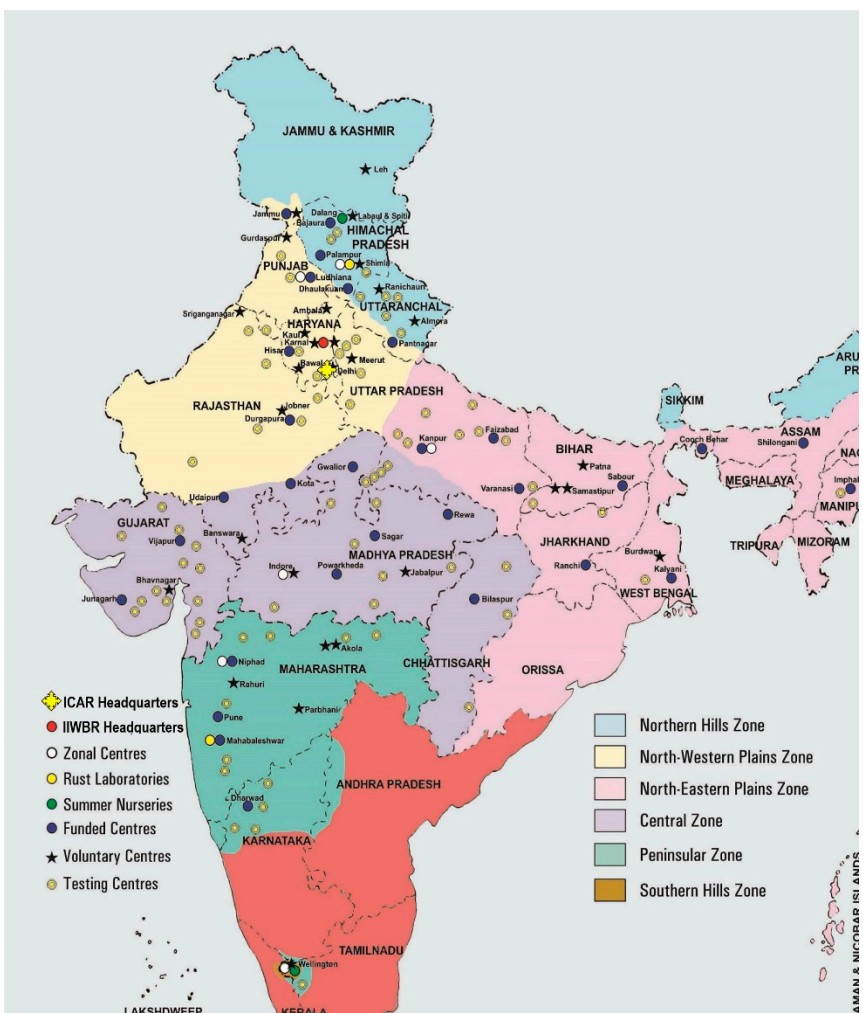

**Figure 1.** Wheat research Institutes and wheat growing zones of India.

## 2. Status of Wheat Rust Research in India

Wheat is grown under diverse climatic conditions in India. All three wheat rust pathogens are prevalent both in regular as well as summer crops. Wheat rust research in India started around 1922, with the earliest pathotype identification being documented in 1931 [3]. The basic objectives of the research conducted were wheat rust surveillance, identification of pathotypes, understanding the epidemiology of the rust pathogens of wheat and barley, and the identification of rust resistance sources in wheat that led to breeding for rust resistance [4]. Over the years, research activities on rusts

have changed to address new challenges that have arisen due to virulence shifts in rust pathogen populations. In current research efforts, emphasis is laid on the regular monitoring/pathotyping of wheat rust pathogens, the use of NILs in the identification of pathotypes, evaluation of rust resistance in germplasm, pre-emptive breeding for rust resistance, and the strategic deployment of wheat varieties with diverse rust resistances. In addition, research activities were initiated targeting the pyramiding of rust resistance genes, investigating genetic variability among wheat rust pathotypes, genome sequencing of wheat rust pathogens, molecular studies of host–pathogen interactions, and revisiting the epidemiology of wheat rust pathogens.

### 2.1. Monitoring of Wheat Rusts in India

Although monitoring of wheat rusts in India began in the 1920s, systematic monitoring of wheat rusts began around 1967. Wheat Disease Trap Plot nurseries have been planted since then to monitor the occurrence of wheat diseases in different parts of India, allowing the occurrence and migration pathways of the wheat pathogens and in particular the rusts to be determined. This was extended to neighboring countries through the establishment of another nursery comprising wheat lines from South Asian Association for Regional Cooperation (SAARC) nations that is planted in neighboring countries also.

### 2.2. Identification of New Pathotypes and Mapping the Distribution of Puccinia Species on Wheat in India

Since 1931, 145 pathotypes of the three wheat rust pathogens have been identified, all of which are maintained in the national repository at Flowerdale, Shimla, India. The distribution and mapping of pathotypes is done regularly [3,5,6]. During this time, there have been some landmark changes in the wheat rust pathogen populations. Virulence within *P. striiformis* for the cultivar Kalyansona first appeared around 1971, followed by virulence for cv. Sonalika between 1984–1990 [6]. In 1996, virulence for *Yr9* was first detected, followed by a pathotype that has combined virulence for *Yr9* and *Yr27* in 2001 [7]. Subsequently, three new pathotypes with additional virulence on Riebesel47/51, SuwonXOmar, *Yr11*, *Yr14*, and *Yr24* have been identified [8]. Pathotype 121R63 of *P. triticina* predominated for more than 20 years, but in 2016 a shift in favor of pathotype 121R60-1 occurred [9]. Many virulent pathotypes have been reported in *P. graminis tritici*, however, little change has occurred in the predominance of pathotypes. Pathotype 62G29 remained common for a long time, however, recently predominance of 79G31 has been documented [10]. The predominant pathotypes of the three wheat rust pathogens in India and their virulence structures are listed in Table 1.

**Table 1.** Predominant pathotypes of *Puccinia* species on wheat in India.

| Rust Pathogen | Predominant pathotypes * | Avirulence/Virulence Structure |
|---|---|---|
| *P. striiformis tritici* | 46S119 = 46E159 | *Yr1, Yr5, Yr10, Yr11, Yr12, Yr13, Yr14, Yr15, Yr16, Yr24, Yr27, YrSp, YrSD/Yr2, Yr3, Yr4, Yr6, Yr7, Yr8, Yr9, Yr17, Yr18, Yr19, Yr21, Yr22, Yr23, Yr25, YrA, YrSD, YrSo* |
| | 110S119 = 110E159 | *Yr1, Yr5, Yr10, Yr13, Yr14, Yr15, Yr16, Yr24, Yr26, YrSp, YrSk/Yr2, Yr3, Yr4, Yr6, Yr7, Yr8, Yr9, Yr11, Yr12, Yr17, Yr18, Yr19, Yr21, Yr22, Yr23, Yr25, YrA, YrSo* |
| *P. graminis tritici* | 79G31 = RRTSF (11) | *Sr7a, Sr8a, Sr8b, Sr9e, Sr22, Sr23, Sr24, Sr25, Sr26, Sr27, Sr31, Sr32, Sr33, Sr35, Sr37, Sr39, Sr40, Sr43, SrTmp, SrTt3/Sr2, Sr5, Sr6, Sr7b Sr 9a, Sr 9b, Sr 9c, Sr 9d, Sr 9f, Sr9g, Sr10, Sr11, Sr13, Sr14, Sr15, Sr16, Sr17, Sr18, Sr19, Sr20, Sr21, Sr28, Sr29, Sr30, Sr34, Sr36, Sr38, SrMcN* |
| | 62G29 = PTHSC (40A) | *Sr7a, Sr13, Sr21, Sr22, Sr24, Sr25, Sr26, Sr27, Sr30, Sr31, Sr32, Sr33, Sr35, Sr36, Sr37, Sr39, Sr40, Sr43, SrTmp, SrTt3/Sr2, Sr5, Sr6, Sr7b, Sr 8a, Sr 8b, Sr 9a, Sr 9b, Sr 9d, Sr 9e, Sr 9f, Sr 9g, Sr10, Sr14, Sr15, Sr16, Sr17, Sr18, Sr19, Sr20, Sr23, Sr28, Sr29, Sr34, SrMcN* |

**Table 1.** *Cont.*

| Rust Pathogen | Predominant pathotypes * | Avirulence/Virulence Structure |
|---|---|---|
| *P. triticina* | 121R60-1 = MHTKL (77-9) | *Lr2a, Lr 2b, Lr2c, Lr9, Lr19, Lr24, Lr25, Lr28, Lr32, Lr 39, Lr42, Lr45, Lr47/Lr1, Lr3, Lr10, Lr11, Lr12, Lr13, Lr14a, Lr14b, Lr14ab, Lr15, Lr16, Lr17a, Lr17b, Lr18, Lr20, Lr21, Lr22a, Lr22b, Lr23, Lr26, Lr27+Lr 31, Lr30, Lr33, Lr34, Lr35, Lr36, Lr37, Lr38, Lr44, Lr46, Lr48, Lr49* |
| | 121R63-1 = THTTM (77-5) | *Lr9, Lr19, Lr24, Lr25, Lr28, Lr29, Lr32, Lr39, Lr42, Lr43, Lr45, Lr47/Lr1, Lr2a, Lr2b, Lr2c, Lr3, Lr10, Lr11, Lr12, Lr13, Lr14a, Lr14b, Lr14ab, Lr15, Lr16, Lr17a, Lr17b, Lr 18, Lr20, Lr21, Lr22a, Lr22b, Lr23, Lr26, Lr27, Lr30, Lr33, Lr34, Lr35, Lr36, Lr37, Lr38, Lr40, Lr44, Lr48, Lr 49* |
| | 21R55 = PHTTL (104-2) | *Lr9, Lr10, Lr13, Lr15, Lr19, Lr20, Lr24, Lr25, Lr28, Lr29, Lr32, Lr36, Lr40, Lr41, Lr42, Lr43, Lr45/Lr1, Lr2a, Lr2b, Lr2c, Lr3, Lr11, Lr12, Lr14a, Lr14b, Lr 14ab, Lr16, Lr17a, Lr18, Lr21, Lr22a, Lr22b, Lr23, Lr 26, Lr27+31, Lr30, Lr33, Lr34, Lr35, Lr37, Lr38, Lr Lr44, Lr48, Lr49* |

* European equivalent of *P. striiformis tritici* pathotypes [11], North American equivalents for *P. graminis tritici* [12], and *P. triticina* [13].

Over the years many resistance genes like *Lr9, Lr19, Lr26, Lr28, Sr24, Sr25;* and *Yr9* have been overcome by the development of matching virulences. As a part of a proactive program, detection of new pathotypes in initial stages, identification of resistant material, pre-emptive breeding, and deployment of resistant wheat varieties are carried out [14].

### 2.3. Epidemiology of Wheat Rusts in India

Under Indian conditions, alternate hosts do not play any role in the perpetuation or epidemiology of wheat rust pathogens. The non-synchronization of vulnerable tender barberry leaves with the availability of basidiospores, drastically, curtails the role of alternate hosts under Indian conditions. Although plenty of teliospores of *P. graminis* f. sp. *tritici* are formed on wheat in the plains region, *Berberis* species occur only in the hills. Moreover, wheat harvest is succeeded by hot summers, and rains which do not leave any chances of survival of obligate parasite rust in absence of wheat. In fact, alternate hosts are of no consequence in the recurrence of stem rust in India [3,5,15,16]. So far four types of aecial stages have been recorded on *Berberis* that are connected with hosts other than wheat:

a. An aecial stage commonly present in Shimla hills has been connected with *Agropyron semicostatum* Nees and is *P. graminis* f. sp. *agropyrii* [17].
b. Aecial stage on *Berberis jaeschkeana* C.K. Sch. has been connected with *Poa nemoralis* and named as *P. poae-nemoralis* Otth. [18].
c. The third aecial stage of *Berberis* in Shimla is connected with *Brachypodium sylvaticum* (Huds.) Beauv. being *P. brachypodii* [19].
d. *Aecidium montanum* Butler which is frequently observed in Shimla, Kumaon hills and Nepal could not be related to *P. graminis tritici* and has been retained as *Aecidium montanum*.

The survival and perpetuation of the wheat rust pathogens in India is believed to occur on self-sown wheat plants and summer crop in the form of uredospores in the hills. Many collateral hosts such as *Aegilops squarrosa, A. ventricosa, A. trinecilis, Bromus carinatus, B. coloratus, B.japonicus, B. mollis, B. patulus, Hilaria jamesii, Hordeum distichum, H. murinum, H. stenostachys, Lolium perenne,* and summer sown wheat in higher hills, self-sown and volunteer plants of wheat etc. also serve as offseason hosts in the annual recurrence of the stem rust fungus.

A similar situation holds true for other rust pathogens also. Alternate hosts of leaf rust, especially *Thalictrum* species, are common in the hills but aecial structures have not been found on these. Likewise, *Berberis* species occurring in hills were not found to harbor any *P. striiformis* aecial cups. It is assumed that the rust fungi have shed their alternate hosts and are known to be evolutionary more advanced than those which have functional alternate hosts [14,16].

We conducted detailed studies on the role of *Berberis* in the recurrence of wheat rusts in India. Aeciospores collected from the leaves of seven species naturally growing in Northern India and Nepal were used to infect wheat by following the procedure of Jin and coworkers [20]. These studies failed to induce any infection on wheat. To further monitor the role of hills in the spread of rusts to the plains, we planted Wheat Disease Trap Plot Nurseries at 10 locations in the Northern hills zone and the North Western Plains zone in the off-season (Figure 1). Apart from Shimla where the plots were protected, the entries of wheat disease nursery did not survive in the rainy season at other locations. Leaf rust alone was observed on this nursery at Shimla. These studies indicated that rusts do not survive on either *Berberis* species or on wheat in the offseason. Hence, it appears that in India wheat rusts survive on some collateral hosts or alternative hosts or unknown alternate hosts [14]. We have observed rust on some grasses, which do not infect wheat. Rust on these grasses appeared twice in the year when conditions were favorable, however, the rust was dormant during extreme summers and winters. We could get some rust infection on Congress grass (*Parthenium hysterophorus*). Its inoculation on wheat yielded leaf rust infection and we could prove Koch's postulates also. Pathotype analysis revealed a mixture of pathotypes and one of those as a new one (Kumar et al. under publication). Congress grass is a noxious weed and is common in many areas [21].

## 2.4. Race Ug99 of P. graminis f. sp. tritici

Historically, stem rust is a most damaging disease of wheat. Although well-controlled for many years, this disease reemerged as a threat with the detection of Ug99, a race with virulence on *Sr31* that was first isolated from Uganda in 1998 [22]. Dr. Norman Borlaug led the call for a joint effort to tackle this threat, which led to the establishment of the Borlaug Global Rust Initiative (BGRI, earlier Global Rust Initiative) in September 9, 2005 at Nairobi, Kenya with the following objectives: To monitor the movement of race Ug99, to evaluate released varieties and germplasm for resistance to Ug99, to share sources of stem rust resistance worldwide, and to incorporate diverse resistance and adult plant resistance into high-yielding adapted wheat varieties. Under the framework of the BGRI, the evolution and migration route of the Ug99 group of races are being monitored carefully so as to provide early warning to all stakeholders in case of an epidemic. Over the years, 13 pathotypes have been identified within the Ug99 lineage, one or more of which have been detected in 13 countries [23]. India, one of the partners of the BGRI, is actively participating in germplasm screening in Kenya and Ethiopia. A Global Cereal Rust Monitoring System (GCRMS) has been implemented under the umbrella of BGRI, Consultative Group on International Agricultural Research (CGIAR) centres, advanced research labs, national agricultural programs and UN-FAO, to integrate and disseminate up to date information on stem rust incidence as well as the distribution of pathotypes. The GCRMS has resulted in the emergence of a strong, rapidly expanding, coordinated international rust surveillance network. So far more than 15 countries are reporting standardized field survey and surveillance data on wheat rust disease incidence and severity, and this number is expected to rise further in the near future [24]. As preparedness for combating the probable threat of Ug99, more than 250,000 wheat genotypes including advanced breeding materials from wheat producing countries of Africa and Asia were screened for resistance to stem rust race Ug99 and its derivatives at Njoro, Kenya and to a lesser extent, at Kulumsa and Debre Zeit, Ethiopia from 2005 to 2012 [25]. The efforts of the BGRI, CIMMYT, and Kenyan Agricultural and Livestock Research Organisation and the Ethiopian Institute of Agricultural Research have been very successful in training many people and in identifying resistant germplasm. At present, a stem rust prone area of about 7 million hectares in Central and Peninsular India has been planted to Ug99 resistant germplasm.

Our pathotype analyses have revealed that pathotypes within the Ug99 lineage do not occur in India or in neighboring countries, and that *Sr31* is still effective against the stem rust pathogen in India [10]. Moreover, India has a proactive system to combat these threats. Despite the absence of this group of stem rust pathotypes in India, we remain vigilant and ongoing rust surveillance, pathotype monitoring and varietal deployment is in place and there is effective preparedness to any rust threat to wheat.

### 2.5. Pyramiding of Rust Resistance Genes and Deployment of Genetic Stocks

Although different rust resistance genes are available in various backgrounds, in many cases useful resistance is associated with genetic drag. Because rust resistance genes such as *Lr9, Lr19, Lr26,* and *Yr9* have been rendered ineffective, other genes are needed in desirable backgrounds to combat wheat rusts [6]. Moreover, there is always a need to have diverse and pyramided resistance available for use by breeding programs. To overcome these challenges, an extensive program was initiated at Shimla to pyramid useful resistance genes in acceptable agronomic backgrounds [6]. Wherever required, marker assisted confirmation of resistance is used. Consequently, 35 rust resistant genetic stocks were developed and registered with National Bureau of Plant Genetic Resources (NBPGR), New Delhi (Table 2).

**Table 2.** Genetic stocks with pyramided rust resistance in good agronomic background.

| S. No. | Title (INGR Number) | Year of Notification | Resistance to Wheat Rusts | Remark/Resistance Gene |
|---|---|---|---|---|
| 1. | FLW1 (03013) | 2003 | Resistant to leaf and stem rusts | *Lr24 + Sr24* |
| 2. | FLW2 (03014) | 2003 | Resistant to leaf and stem rusts | *Lr24 + Lr26 + Sr24 + Sr31 + Yr9 + Yr27* |
| 3. | FLW3 (03015) | 2003 | Resistant to stripe and stem rusts | *Lr26 + Sr31+ Yr9+ YrChina-84* |
| 4. | FLW4 (03016) | 2003 | Resistant to leaf and stem rusts | *Lr24+ Lr26+ Sr2+ Sr24+ Sr31+ Yr9* |
| 5. | FLW5 (03017) | 2003 | Resistant to leaf and stem rusts | *Lr24+ Sr24* |
| 6. | FLW6 (04011) | 2003 | Resistant to leaf and stem rusts | *Lr9+ Lr24+ Sr2+Sr24* |
| 7. | FLW8 (04012) | 2004 | Resistant to leaf and stem rusts | *Lr19+ Sr25* |
| 8. | HW2002(04014) | 2004 | Resistant to leaf and stem rusts | *Lr24/Sr24* |
| 9. | HW2031(04015) | 2004 | Resistant to all the rusts | *Lr28* |
| 10. | HW2049 (04016) | 2004 | Resistant to all the rusts | *Lr19/Sr25* |
| 11. | FLW11 (05003) | 2005 | Resistant to stripe and stem rusts | *Lr26+ Sr31+ Yr9+ YrHobbit* |
| 12. | FLW12 (05004) | 2005 | Resistant to stripe and stem rusts | *Lr26 + Sr31 + Yr9 + YrMega* |
| 13. | FLW13 (05005) | 2005 | Resistant to stripe and stem rusts | *Lr34 + Sr2 + Yr15 + Yr18* |
| 14. | FLW15 (05006) | 2005 | Resistant to leaf and stem rusts | *Lr26 + Lr32 + Sr31 + Yr9 + YrPBW343* |
| 15. | RNB1001 (05007) | 2005 | Resistant to leaf and stripe rusts | *Lr unknown, Yr unknown* |
| 16. | FKW1 (06004) | 2006 | Resistant to leaf and stem rusts | *Lr26 + Sr31 + Yr9 + YrChina-84* |
| 17. | FKW3 (06005) | 2006 | Resistant to leaf rust | *Lr unknown* |
| 18. | FLW20 (07001) | 2007 | Resistant to leaf and stem rusts | *Lr26 + Lr19 + Lr24 + Sr31 + Yr9 + YrPBW343* |
| 19. | FLW24 (07005) | 2007 | Resistant to leaf and stem rusts | *Lr19 + Lr26 + Sr25 + Sr31 + Yr9 + Yr27* |
| 20. | FLW25(07006) | 2007 | Resistant to leaf and stem rusts | *Lr28 + Lr26 + Sr31 + Yr9 + Yr27* |
| 21. | FLW26 (07007) | 2007 | Resistant to leaf and stem rusts | *Lr42 + Lr26 + Sr31 + Yr9 + Yr27* |
| 22. | FLW27 (07008) | 2007 | Resistant to leaf and stem rusts | *Lr45 + Lr26 + Sr31 + Yr9 + Yr27* |
| 23. | FLW28 (08001) | 2008 | Resistant to leaf and stripe rusts | *Lr19 + Lr24 + Lr26 + Sr24 + Sr25 + Sr31 + Yr9* |
| 24. | FLW29 (08002) | 2008 | Resistant to all the rusts | *Lr26 + Lr28 + Sr31 + Yr9 + YrCD* |
| 25. | FLW30 (08003) | 2008 | Resistant to all the rusts | *Lr26 + Lr28 + Sr31 + Yr9 + Yr15* |
| 26. | DWRL-1 (12019) | 2012 | Resistant to all the rusts | - |
| 27. | FLW10(INGR17006) | 2017 | Resistant to leaf and stripe rusts | *Lr26 + Sr31 + Yr9 + Yr10* |
| 28. | FLW16(INGR17007) | 2017 | Resistant to leaf and stripe rusts | *Lr26 + Sr31 + Yr9 + Yr5* |
| 29. | FLW21(INGR17008) | 2017 | Resistant to all wheat rusts | *Lr26 + Lr24 + Sr24 + Sr31 + Yr9 + Yr15* |
| 30. | FLW22(INGR17009) | 2017 | Resistant to all wheat rusts | *Lr26 + Lr28 + Sr31 + Yr9 + YrChina-84* |
| 31. | FWW2(INGR17010) | 2017 | Resistant to leaf, stem rusts and Ug99 type virulences | *Lr19 + Lr24 + Lr26 + Sr24 + Sr25 + Sr31 + Yr9* |
| 32. | FLW31(INGR17040) | 2017 | Resistant to leaf and stem rusts and Ug99 group of pathotypes | *Sr24/Lr24 and Sr43* |
| 33. | FLW32(INGR17041) | 2017 | Resistant to stem rust and Ug99 type of pathotypes. | *Lr10 + Lr13 + Sr2 + Sr26+ Yr2KS + Yr5* |
| 34. | FLW33(INGR17042) | 2017 | Resistant to leaf and stem rusts and Ug99 group of pathotypes. | *Sr24/Lr24 and Sr32* |
| 35. | FLW18(INGR17070) | 2017 | Resistant to leaf rust | *Lr26 + Lr39 + Sr31 + Yr9 + Yr (PBW343)* |

These diverse genetic stocks have been made available for use by different wheat breeding programs to develop rust resistant wheat in India ([6], Personal comm, S.C. Bhardwaj).

### 2.6. Breeding for Rust Resistance in Wheat

Systematic breeding for rust resistance in wheat in India began in the early 1950s. Much has been achieved through the years in controlling rusts by planting resistant wheat cultivars with diverse resistance sources. Such genetic diversity has not only proved critical in developing rust resistant cultivars but also in understanding rust epidemiology, and has gradually reduced the quantum and frequency of wheat rust epidemics. The resistance gene *Lr26* in combination with *Lr13, Lr23,* and *Lr34* and the *Agropyron* segment carrying *Lr24/Sr24* have played a crucial role in providing field resistance and protecting wheat from any leaf rust epidemic threat to sustained wheat production. Likewise *Sr31* in combination with *Sr2, Sr24, Sr5,* and *Sr8* have provided protection against stem rust whereas *Yr9* in combination with *Yr2, Yr18,* and some unknown adult plant resistance genes conferred protection from stripe rust. In recent years, wheat has achieved relatively higher production stability as compared to other cereal crops by adopting strategic varietal deployment. Only marginal increases in the wheat area is recorded but the planned deployment of rust resistant wheat varieties is the most viable tool of crop protection and crucial in sustaining the production levels. Detailed accounts of Indian efforts in breeding wheat for disease resistance are available (e.g. [26]). Marker assisted backcross breeding has become an integral part of Indian wheat breeding programs [6,14,27]. In recent years, five new varieties with resistance to all three rusts have been recommended for cultivation through this technique.

### 2.7. Screening of Wheat Germplasm for Rust Resistance

All wheat germplasm developed under the aegis of the All India coordinated Wheat and Barley Improvement Project are subjected to rigorous screening for seedling (all stage), race specific Adult Plant resistance (APR) and non-race specific resistance at the Indian Institute of Wheat and Barley Research, Regional Station, Flowerdale, Shimla, Himachal Pradesh (Figure 1). The screening involves challenging each wheat line against most virulent predominant pathotypes with different avirulence/virulence structures. Studies involve screening wheat material against 70 different pathotypes of the three rust pathogens under controlled temperature and light conditions. In addition, the genetic basis of rust resistance is also investigated. Based on the gene matching technique, rust resistance genes are postulated in wheat material. A list of rust resistance genes so inferred in Indian wheat material is given below Table 3:

**Table 3.** Diversity for rust resistance genes in identified cultivated wheat and material in pipeline.

| Rust | Number of Genes Inferred: Details of Resistance Genes |
|---|---|
| Stripe | Six: *Yr2, Yr9, YrA, Yr18, Yr29,* and *Yr46* |
| Leaf | Nineteen: *Lr1, Lr2a, Lr3, Lr9, Lr10, Lr13, Lr14a, Lr17, Lr18, Lr19, Lr22, Lr23, Lr24, Lr26, Lr*28, *Lr34, Lr46, Lr49,* and *Lr67* |
| Stem | Twenty-two: *Sr2, Sr5, Sr6, Sr7*a, *Sr7b, Sr8a, Sr8b, Sr9b, Sr9e, Sr11, Sr12, Sr13, Sr17, Sr21, Sr24, Sr25, Sr28, Sr30, Sr31, Sr55, Sr57,* and *Sr58* |

To identify slow rusting and APR to rusts, advanced varietal trials or pipeline wheat material is also evaluated at hotspot locations by creating artificial epiphytotics. Simultaneously race specific resistance is also explored by using specific virulent and most predominant pathotypes of each rust pathogen [28].

### 2.8. Deployment of Wheat Varieties

Rust resistance is the most economic, ecologically safe and effective way to manage wheat rusts. There are always regional differences in the distribution of rust pathogens, as well as of the pathotypes of each rust pathogen. Based on the distribution of pathotypes of the different *Puccinia* species on wheat and the rust resistance of wheat varieties, deployment of rust resistant wheat varieties is undertaken tactfully in different wheat growing areas [14,27]. A combination of all stage (seedling) resistance, slow rusting resistance, and APR of both the race specific and non-race specific types is deployed for the

strategic management of wheat rusts [27,28]. Diversity for rust resistance within an area has become the key factor for managing wheat rusts in India.

## 2.9. Gene Mining

Efforts to search for new and unexplored rust resistance genes are ongoing, targeting present day wheat material, local germplasm, and exotic accessions [6]. Two examples of resistance genes identified in this way in India are *Lr48* and *Lr49* [29]. Recently, a new leaf rust resistance gene was mapped to chromosome 2DS. This gene confers resistance to leaf rust and probably also stem rust (Subodh Kumar, personal communication). Gene mining work for leaf rust resistance was also undertaken in a set of Indian wheat germplasm [30]. Quantitative Trait Loci for leaf rust resistance (six) and for stripe rust resistance (five) have been investigated in wheat material by using Indian isolates of two rust pathogens [31].

## 2.10. DNA Polymorphism among Wheat Rust Pathogens/Pathotypes

Molecular analysis of Indian pathotypes of *Puccinia triticina* based on SSR markers indicated high genetic diversity and the presence of seven major genotypic clusters. These findings offer valuable information for framing suitable disease management strategies through appropriate region-specific gene deployment, and improve the understanding of the population biology and evolution of this pathogen in the Indian subcontinent [32]. Similar studies have been conducted for other wheat rust pathogens also [8,10].

## 2.11. Sequencing of Wheat Rust Genomes

To understand the molecular basis of variability in rust fungi, the genome of *P. triticina* was decoded. A comparison was made between the highly variable Race 77 and its 13 biotypes and the stable Race 106. Race 106 was first identified in 1930, is preserved in the National collection at Shimla and has not evolved during the past more than 89 years, whereas race 77 first detected in 1954 from Pusa (Bihar), has evolved 13 biotypes affecting wheat breeding in the country. Therefore, it was very important to understand the molecular mechanism for the virulence and adaptation that has been documented within Race 77, and unravelling the molecular basis of its fast evolution and stability of genome of race 106 genome. At Indian Council of Agricultural Research-National Research Centre for Plant Biotechnology (NRCPB), New Delhi (Figure 1), Next generation sequencing (NGS) technology was used to decode the genomes of 15 races (~1500 Mb data) of *P. triticina*. A high-quality draft genome (~100Mb) sequence of Race 77 with 33X genome coverage was generated, and 27,678 protein coding genes responsible for various functions were predicted. Genome wide comparative analysis revealed that the *P. triticina* genome is 37% and 39% repetitive in case of Race 77 and Race 106, respectively and Race 77 substantially differs from Race 106 at segmental duplication (SD), repeat elements and SNP/InDel levels. We found certain "hot spot regions" in the genomes of Race group 77 which are vulnerable for reshuffling, leading to variability in it. This study provides insight into *P. triticina*, with emphasis on the genome structure, organization, molecular basis of variation and pathogenicity. This genomic information was an important landmark research in India and will facilitate wheat improvement in the country [33]. Likewise, the genome of *P. striiformis tritici* (wheat stripe rust fungus) was also decoded [34]. The genomes of four Indian pathotypes of *P. graminis tritici* have also been decoded (Kanti Kiran, personal communication).

## 2.12. Molecular Bases of Host Pathogen Interaction

Studies were initiated on wheat-rust pathogen interactions at the molecular level. Temporal expression pattern of defense responsive genes during early infection phases in compatible and incompatible interactions between wheat and *P. triticina* were compared to elucidate the molecular basis of *Lr24* - mediated resistance. *Lr24* confers resistance to all pathotypes of *P. triticina* reported from the Indian subcontinent. The expression profiles of these genes revealed that the transcript levels

of aquaporin, endochitinase, β-1,3-glucanase, Type 1 non-specific lipid transfer protein precursor, phenylalanine ammonia-lyase, and caffeic acid O-methyltransferase were significantly higher in the incompatible interaction. A relatively higher expression of phenylalanine ammonia-lyase and aquaporin was observed at the pre-haustorial stage (6 h post inoculation (hpi) and 12 hpi), while expression of endo-chitinase and lipid transfer protein was higher during the post-haustorial stage (24 hpi and 48 hpi) in the incompatible interaction. Induced expression of class III peroxidase was maintained throughout pre- and post-haustorial stages in the compatible interaction. The results of the current study could distinguish the defense responsive genes playing a critical role in wheat resistance or susceptibility to leaf rust. We feel that this understanding will allow the development of novel strategies to regulate wheat: *P. triticina* interactions through genetic engineering [35].

To understand the role of SA and sugar-mediated resistance mechanisms during early leaf rust infection on wheat, expression profiles of the main regulators of SA (TaEDS1, TaPAD4, TaNDR1, TaRAR1, TaSGT1, TaHSP90, TaEDS5, TaPAL, and TaNPR1) and sugar (TaHTP, TaSTP13A) pathways were analyzed between two wheat near-isogenic lines (NILs) with *Lr24* and without *Lr24*. The quantification of candidate gene expression using reverse transcription quantitative real-time PCR at different time points post inoculation revealed stage-specific transcriptional reprogramming between susceptible and resistant expressions. The genes acting upstream of SA in the SA pathway (TaEDS1, TaPAD4, TaNDR1, TaRAR1, TaSGT1, TaHSP90, and TaEDS5) exhibited strong expressions at a later stage (48 h post inoculation (hpi)) of leaf rust infection in the susceptible response compared to unchanged or slightly changed expressions in the resistant interaction. On the other hand, the genes involved in SA biosynthesis (TaPAL), SA downstream signaling (TaNPR1), and sugar transportation (TaHTP, TaSTP13A) exhibited a strong expression at mid phase of infection between 6 and 24 hpi in the resistant compared to the susceptible response. These expression patterns suggest that TaPAL and TaNPR1 play a positive regulatory role in the SA-mediated resistance pathway whereas TaHTP (*Lr67*) plays an important role in the sugar-mediated resistance pathway activated by the leaf rust resistance gene, *Lr24* [36]. Similar studies are underway for the stem and stripe rust pathogens also.

## 3. Success in the Management of Wheat Rusts in India

Wheat is cultivated on about 30 million hectares in India. A wide variety of wheats including bread, durum and monococcum are grown. Owing to variable climatic conditions for growing wheat, all the wheat rust pathogens are prevalent in India. In spite of all the above facts, wheat rusts have been contained successfully for the last 47 years, while wheat production has kept on growing linearly reaching 102 million tons mark and there has been a saving of 10 million tons every year through successful management of wheat rusts. Moreover, chemicals were hardly used to contain the initial foci of infection.

A vigil is kept on the occurrence of wheat rusts in India through the collaborative and coordinated efforts of wheat workers. Pathotype mapping and screening for rust resistance is a regular feature of the Indian wheat program with the activities undertaken at Indian Wheat and Barley Research Institute, Regional Station, Flowerdale, Shimla, Himachal Pradesh. The centre has a collection of 145 pathotypes of different rust pathogens and is responsible for providing the nucleus and bulk inocula for different research centres elsewhere in India. The main strategy has been to deploy wheat varieties with diverse rust resistance by using the pathotype distribution data skillfully [37]. A combination of all-stage resistance, slow rusting and APR (race specific and non-race specific types) is used Anticipatory rust resistance breeding has always been the backbone effort of the wheat programs. An emphasis is laid on the diversity of the wheat varieties grown in an area. [38]. A challenge is posed by the cultivation of mega varieties over a large area, e.g. PBW343. When PBW343 became susceptible to stripe rust, it was replaced expeditiously with many varieties within three years. This was possible because a number of choices for wheat varieties were available for farmers. Wheat rust management in India has been a model decision support system in management of diseases in agriculture.

## 4. Challenges Ahead

Alternate hosts of wheat rusts like *Berberis* species are nonfunctional under Indian conditions. Like many other countries (Australia, New Zealand, South Africa, Ecuador, Pakistan, Nepal, Bangladesh, and Bhutan) in the world, wheat rust populations in India multiply clonally by producing uredospores. However, wheat rusts being obligate pathogens, require a living host for their continuance. Earlier, it was hypothesized that rusts survive on bridging wheat crop, volunteer/self-sown plants in hills and provide inoculum for the plains. However, now a days there is little wheat grown in the hills, and rusts appear in the plains of India earlier than the hills in usual wheat season. Therefore, the epidemiology of wheat rusts would be re-investigated with the changed situations. While we are comfortable with resistance to leaf and stem rusts in wheat in India, there is a need for diversifying stripe rust resistance. Recently efforts in this direction were fruitful with the identification of wheat varieties with entirely different types of resistance to stripe rust.

### 4.1. Present Scenario

At present, there is practically no wheat cultivation in the off-season in the hills, with the exception of the Indian Agric. Research Institute, Wellington (Tamil Nadu), Indian Institute of Wheat and Barley Research, Dalang Maidan (Lahaul and Spiti), Kinnaur (H.P.) and Leh Ladakh. Our investigations for many years have revealed that the wheat rust flora of the Leh Ladakh and Kinnaur area is very primitive, comprising races that do not occur elsewhere in India. Thus, this excludes the possibility that Leh Ladakh and Kinnaur play any role in the epidemiology of wheat rusts in India [39]. Moreover, new pathotypes of wheat stripe rust first appear in Ropar, Gurdaspur and other areas in Punjab and recorded in Himachal Pradesh including Lahaul and Spiti after 5–6 years as has been observed in case of *Yr9* virulences 46S119 and 78S84 earlier and 110S119 recently [6,7,16].

### 4.2. Evolution of Wheat Rust Pathogen in India over the Years?

In countries where rusts multiply clonally, evolution of new pathotypes may be through introduction from other countries or may be brought about by the changes in the local population [6,14,16]. Earlier simple races of *Puccinia striiformis* used to exist in India. With the introduction of dwarf wheat varieties like Kalyansona and Sonalika, the pathogen flora also changed. Cultivation of rust resistant wheat also favors selection of new mutants on resistant wheat varieties. With the evolution of new races, resistance of these wheat varieties was overcome. Generally new pathotypes evolve through mutation but somatic hybridization has also been observed in a few cases [6,27]. In most of the cases there had been a stepwise gain in virulence for one resistance gene, however, reverse mutation may be the cause of new variation as has been observed for a few resistance genes like *Lr2a, Lr2c, Lr20, Sr11* and *Yr1* [6,14].

## 5. Conclusions

The rust pathogens of wheat are devastating as they evolve continuously, and their uredospores spread by air over long distances. This has resulted in failure of crop resistance and severe losses to wheat production. As an emergent tool for managing wheat rusts in India, fungicides belonging to triazole (Propiconazole, Tebuconazole, Triadimefon), are kept ready for effectively controlling wheat rusts at the rate of 0.1 percent. However, resistant cultivars have remained popular among the farmers as they are cost effective and environmentally neutral in terms of impact. Further, growers are boosted to adopt rust resistant cultivars. A combined strategy including regular disease surveys, strengthening research capacity, development of new rust resistant varieties and ensuring their adoption, have led to effective management of rust diseases and boost wheat production in India. The claim is well verified with the fact that India has had no wheat rust epidemic for the last 47 years even when many countries in the world have had rust outbreaks.

**Author Contributions:** S.C.B., O.P.G. drafted and edited the manuscript, G.P.S., P.P., and S.K. compiled data and prepared tables. All authors have read and approved the final manuscript.

**Funding:** This research received no external funding.

**Acknowledgments:** We acknowledge the receipt of funding from Department of Biotechnology and Indian council of Agriculture Research, New Delhi, India for undertaking the different activities of research presented in this manuscript. We are grateful toRobert Park, Judith and David Coffey Chair of Sustainable Agriculture, Faculty of Science, School of Life and Environmental Sciences, Plant Breeding Institute, The University of Sydney, Australia for critical appraisal of the manuscript and improving it thoroughly.

**Conflicts of Interest:** The authors declare no conflict of interest.

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
