# Peer review of "Status of Wheat Rust Research and Progress in Rust Management-Indian Context"

_agronomy, doi:10.3390/agronomy9120892_

Round 1
Reviewer 1 Report
The manuscript has significantly been improved for the English style and for the presentation of the different sections.
Author Response
Response is appended along with please.

Reviewer 2 Report
The manuscript has been substantially improved. However, there are still important points to address:
While the title focuses on the status of wheat rust research and management in Indian, the abstract mentions nothing about the status of Indian wheat rust research and management.
The English language and style must still be improved.
Abbreviation in the manuscript: in my previous review of this manuscript, I have indicated that abbreviations are being used in the manuscript without proper definitions. The authors removed the abbreviations I mentioned as an example, but did not address others abbreviations.
Lack of references throughout the manuscript. For example, the statement in line 96 need to be supported by relevant references.
Author Response
Response to Reviewer II is appended please.

This manuscript is a resubmission of an earlier submission. The following is a list of the peer review reports and author responses from that submission.
Round 1
Reviewer 1 Report
Summary of Review for “Status of Wheat Rust Research and Success in its Management-Indian Context”
This study provides a review of the current status of wheat rust research and success in its management: Indian context. Given the fact that wheat rusts are the most damaging diseases of wheat that can cause significant yield losses, the initiation to review on the status of research on wheat rust is very important. However, this review manuscript needs to be revised to provide a comprehensive context of one the most highly studied subject. Some of my major concerns are as follows:
a. The manuscript needs extensive editing in terms of English language, style and content. It is particularly important to mention that wheat rust is the most highly researched patho-system particularly in response to the emergence new aggressive races including Ug99. The global community including the CGIAR center, national agricultural research system (including the of India), the BGRI project facilitated extensive global effort to contain the spread of the new wheat rust races and generally the management of wheat rust diseases. The review could not able to provide a comprehensive review of this development over the last 10 years, particularly considering the fact that India has been an active member of these efforts
b. Some of the topics (titles and sub-titles) are very generic lacking details of the current advances in that particular topic. For example, the subtitle “Breeding for rust resistance in wheat” is one case in point, where the authors fail to provide the details the current breeding effort in India, particularly related how the current advancement in DNA sequencing technologies unfolds to provide opportunities for the application genomic-assisted breeding for rust resistance breeding.
c. Similarly, gene discovery, mapping and cloning has been an important aspect of the effort to manage wheat rust. Extensive efforts have been made both in India and globally and several new genes and QTL have been mapped and characterized, the detail of which is missing in the review.
d. References to the information used in the manuscript are lacking. For example, what is source of the information used in Table 2
e. Abbreviation are used throughout the manuscript without proper definitions. For example, what do “IWBR regional Station”, AVT lines etc ? ,
f. The contents of the manuscript (title and sub-titles) are not well organized. For example, under the title “3. Success in management of wheat rusts in India, all the three sub-titles are not related to this title (For example Evolution of wheat rust pathogen in India over the years?
g. extensive use of vague words, incomplete sentences and ideas in several places, inappropriate use of punctuation markers etc
Reviewer 2 Report
The authors present a summary of the situation of wheat rusts in India over the years, however, the manuscript in present form presents several flows, which must be addresses before it can be published.
1. Overall, the English language and style must be improved.
Please pay attention on how statements and sentences are linked in a paragraph in order to help the reader to a clear and smooth read.
2. There is an extremely lack of references throughout the manuscript. Particularly, for a “Perspective” type of article, references are the basis for the presentation of the data and they would potentially help to a better and more comprehensive presentation of the data.
3. Line 16-20: in the abstract the authors refer to the deployment of the 1BL/1RS translocation lines, however this is not covered at all throughout the manuscript.
4. 2.2. Epidemiology of wheat rusts in India: The authors state that: “Under Indian conditions, alternate hosts do not play any role in the perpetuation and epidemiology of wheat rusts [3,5]” referring to a two very old references. Please provide more actual references if available, otherwise this statement must the rephrased since it is not know the exact role of the alternate host in the epidemiology of this pathogen in India. Line 76-77, 91-92, 93 and 94-95: provide references, which support these statements. These kind of statements need to be supported by relevant references otherwise they must be either omitted.
5. 2.3. Identification of new pathotype and mapping distribution of Puccinia species on wheat in India: It would be helpful for the reader to see to which virulence profile these pathotypes refer to, otherwise, they are not informative.
6. 2.4. Pyramiding of rust resistance genes and deployment of genetic stocks: it would be very informative to add to table 2 when, if any, of these genes or groups of genes were defected.
7. 2.5. Breeding for rust resistance in wheat: the authors refer to the successful breeding strategies used in India, however this paragraph lacks examples of the effectivity of specific genes or group of genes. Please rephrase or merge with previous paragraph (2.4)
8. 2.6. Screening of wheat material for rust resistance: please provide details and/or references about how the pathotypes are virulence characterized, which differentials lines are used, how these are scored, etc.
9. 2.7. Deployment of wheat varieties: this paragraph must be extended and references provided.
10. 2.8. Gene mining and 2.9. DNA polymorphism among wheat rust pathogens/pathotypes: extend and add relevant references to these sections.
11. 2.10. Sequencing of wheat rust genomes: the authors provide only information about Lr genome sequencing. What about Yr and Sr?
12. 2.11. Molecular bases of host pathogen interaction: the authors provide only information about Lr without providing any reference to that. What about Yr and Sr?
13. 3. Success in management of wheat rusts in India: line 214-219: provide references.
14. 3.1. Challenges ahead: Line 235-238: provide references to these statements
Line 254-255: only mutation? Might be other driving forces involve in pathogen evolution. Provide information about that.